# Quantifying the impact of basin dynamics on the regional sea level rise in the Black Sea

Kubryakov A.A.[1,2], Stanichny S.V.[1], Volkov D.L.[3,4]

[1]Federal State Budget Scientific Institution "Marine Hydrophysical Institute of RAS", Sevastopol, Russia

[2]Institute of Earth Sciences, Saint Petersburg State University, St. Petersburg, Russia

[3]Cooperative Institute for Marine and Atmospheric Studies, University of Miami, Miami FL, USA

[4]NOAA Atlantic Oceanographic and Meteorological Laboratory, Miami FL, USA

*Correspondence to*: Kubryakov A.Arseny (arskubr@ya.ru)

**Abstract.** Satellite altimetry measurements show that the magnitude of the Black Sea level trends is spatially uneven. While the basin-mean sea level rise from 1993 to 2014 was about 3.15 mm/year, the local rates of sea level rise varied from 1.5-2.5 mm/year in the central part to 3.5-3.8 mm/year at the basin periphery and over the northwestern shelf and to 5 mm/year in the southeastern part of the sea. We show that the observed spatial differences in the dynamic sea level (anomaly relative to the basin-mean) are caused by changes in the large- and mesoscale dynamics of the Black Sea. First, a long-term intensification of the cyclonic wind curl over the Black Sea observed in 1993-2014 strengthened divergence in the center of the basin and led to the rise of sea level in coastal and shelf areas and a lowering in the basin's interior. And second, an extension of the Batumi anticyclone to the west resulted in ~1.2 mm/year higher rates of sea level rise in the southeastern part of the sea. Further, we demonstrate that the large-scale dynamic sea level variability in the Black Sea can be successfully reconstructed using the wind curl obtained from an atmospheric reanalysis. This allows to correct historical tide gauge records for dynamic effects in order to derive better estimates of the basin-mean sea level change in the past, prior to satellite altimetry era.

## 1 Introduction

The mean sea level (MSL) rise in the Black Sea, as well as in the World Ocean, is mainly caused by the basin's freshwater budget and the thermal expansion of the water column due to warming (Stanev et al., 2000; Goryachkin and Ivanov 2006; Jevrejeva et al, 2006; Cazenave et al., 2010). The relative contribution of different components of the Black Sea level budget has been investigated in a number of earlier studies (e.g. Simonov and Altman, 1991; Stanev et al., 2000, 2002; Peneva et al., 2001; Tsimplis et al., 2004; Goryachkin and Ivanov, 2006; Graek et al, 2010; Ilyin et al., 2012; Volkov and Landerer, 2015; Volkov et al., 2016; Aksoy, 2016). The estimates of the Black Sea level rise over the 20th century, based on tide gauge records, range from 1.5 to 2.5 mm/year (Boguslavsky et al., 1998; Reva, 1997; Tsimplis and Spencer, 1997; Goryachkin and Ivanov 2006), which agrees with ~1.8 mm/year of the global MSL rise during the 20[th] century (Church et al. 2004). Based on satellite altimetry measurements during 1993-2010, the global and the Black Sea MSL then rose at a faster rate of ~3.1 mm/year (Church et al., 2011; Avsar et al. 2015). Both the tide gauge and altimetry records show that sea level trends in the Black Sea are not constant over time (e.g. Goryachkin and Ivanov 2006, Kubryakov and Stanichnyi, 2013): MSL was rising at a very high rate of ~28 mm/year in 1993-1999 (Ducet et al., 1998, Stanev et al., 2000; Cazenave et al,

2002, Goryachkin et al., 2003; Vigo et al., 2005; Yildiz et al., 2008), and then it began to fall by ~3 mm/year in 1999-2007 (Ginzburg et al., 2011).

The basin-wide satellite altimetry measurements have revealed that the Black Sea level change is not uniform, which is related to the dynamic factors that redistribute water within the basin (Stanev et al., 2000, 2001; Korotaev et al., 2001).The main feature of the Black Sea dynamics is the cyclonic Rim current flowing along the continental slope. The general cyclonic circulation results in a lower sea level in the interior of the basin and a higher sea level along the coast (Blatov et al., 1984; Simonov and Altman, 1991, Oguz et al., 1993; Stanev et al., 1990, 2000; Korotaev et al., 2001). It has been shown that the seasonal and interannual variability of the Black Sea circulation is driven by changes in the wind curl averaged over the basin (Blatov et al., 1984; Stanev, 1990, 2000; Korotaev, 2001; Graek et al., 2010; Kubryakov et al., 2016). In winter, the cyclonic wind curl and, therefore, the onshore Ekman transport increase and cause divergence in the center of the basin by moving water towards the coast. The compensating vertical uplift (Ekman suction) in the center of the sea brings cold and saline deep water to the surface, while warm and fresher surface water is pushed towards the coast, where downwelling motions occur (Stanev, 2000, 2004; Korotaev, 2001; Kubryakov et al., 2016). In summer, the cyclonic wind curl weakens, Ekman divergence decreases and the water accumulated along the coast flows back into the basin's interior (Zatsepin et al., 2002; Kubryakova and Korotaev, 2017).

Long-term changes of the Black sea dynamics impact on the spatial heterogeneity of the sea level rise in the basin. Particularly, Vigo et al. (2005) and later Kubryakov and Stanichnyi (2013) showed that the Black Sea coastal sea level is rising 1.5-2 times faster than the sea level in the center of the basin. In this paper, we investigate the spatial structure of the Black Sea level trends, its relation to dynamic processes in the basin and atmospheric forcing. We also explore whether historic tide gauge measurements (prior to satellite altimetry era) can be corrected for dynamic effects in order to obtain better estimates of the basin-mean sea level change in the past.

The sea level rise leads to flooding of low-lying coastal areas, coastal erosion, and as a result, has a negative impact on human activities in the Black Sea coastal zone (Alpar, 2009; Avsar et al., 2016). Coastal erosion has been identified as one of the major problems for the Black Sea beaches (Demirkesen et al., 2009; Kosyan et al., 2012). Estimates show that rise of sea level by 1 cm results in 1-2 metres of coastal erosion (Goryachkin and Ivanov, 2006). Rise of sea level by 50 cm will reduce the area of the Black Sea beaches by approximately 50% (Allenbach et al., 2015). That is why the investigation of the spatial variability of the sea level rise in the Black Sea and its reasons is an important task for the coastal applications.

## 2. Data and Methods

In this study, we used the regional satellite altimetry maps of sea level anomalies (SLA) from Jan 1993 to Dec 2014, produced by Ssalto/Duacs and distributed by Aviso, with support from CNES (www.aviso.oceanobs.com). The maps are based on measurements by up to four satellites and produced on a daily basis with a horizontal grid spacing of 1/8°. The data are routinely corrected for instrumental errors and geophysical effects.

A dynamic atmospheric correction (DAC) is applied to account for the dynamic response of the sea level to atmospheric pressure and wind forcing (Carrere and Lyard, 2003). The DAC combines the high frequencies (periods < 20 days) of a barotropic model of Lynch and Gray (1979) with the low frequencies (periods > 20 days) of the inverted barometer correction, and it significantly reduces the aliasing of the high-frequency sea level variability, especially in coastal regions (Volkov et al., 2007). While it has been suggested that the IB correction may not be necessary in the almost enclosed Black Sea (Ginzburg et al., 2011), a recent study by Volkov et al. (2016) showed

that on the interannual and longer time scales the Black Sea level responds to changes in atmospheric pressure in an inverted barometer manner, i.e. 1 mbar change of pressure corresponds to approximately 1 cm change in sea level.

Over the recent years, a great progress in improving the near-coast measurements has been achieved, which has benefited the regional altimetry products, such as the Mediterranean and Black Sea products. The improvement in the coastal areas of the Mediterranean Sea has recently been demonstrated by Marcos et al. (2015). A reasonable agreement between tide gauge records and near-coast SLA in the Black Sea has also been documented (Volkov and Landerer, 2015; Korotaev et al., 1998; Stanev et al., 2000, 2001; Goryachkin et al., 2003, 2003; Kubryakov et al., 2013; Avsar et al., 2015).

The absolute dynamic topography (ADT) of the Black Sea was computed as the sum of the mapped SLA and a "synthetic" mean dynamic topography of Kubryakov and Stanichny (2011). The zonal and meridional components of the surface geostrophic velocities (ug,vg) were computed from the absolute dynamic topography using geostrophic equations: 
$$u_g = -\frac{g}{f}\frac{\partial h}{\partial y} \quad ; \qquad v_g = \frac{g}{f}\frac{\partial h}{\partial x} \quad ,$$

where h is the absolute dynamic topography; $f$ is the Coriolis parameter; and g is the gravitational acceleration. To describe the basin-scale variability we use the magnitude of geostrophic velocity $U = \sqrt{u_g^{\,2} + v_g^{\,2}}$

The variability of the Black Sea level is decomposed in two parts (e.g. Stanev et al, 2000; Graek et al, 2010): i) the basin-averaged sea level change related to the time-variable amount of water contained in the basin and steric effects and ii) the dynamic sea level (DSL) change due to the redistribution of water within the basin. Because the response of SLA to the low-frequency variability of the Black Sea water budget is almost spatially uniform (Korotaev et al., 2001), the DSL at a particular location (x,y) is defined as the difference between the local ADT(x,y) and the basin-mean sea level, MSL: DSL(x,y)=ADT(x,y) – MSL.

In addition to satellite altimetry data, to compute the wind curl over the Black Sea in 1979-2014, we used the 6-hourly ERA-Interim winds at 10 metres height (Dee et al., 2011). It has been shown that the ERA-Interim winds over the Black Sea coincide well with in-situ meteorological measurements, and describe the variability of the wind direction better than other reanalyses (e.g. MERRA, NCEP, WRF) (Garmashov et al., 2016).

To study the variability of eddy dynamics in the Black Sea, we used an automated "winding angle" (Chaigneau et al., 2008) eddy identification method, described in detail in Kubryakov and Stanichny (2015a,b). For each eddy, the method defines its radius and maximum orbital velocity. At each grid point, it also defines the frequency of eddy observation, i.e. the fraction of the total time when the grid point is located within an eddy. Because the Black Sea anticyclones are larger and more powerful than cyclones (Oguz et al, 1993; Kubryakov and Stanichny (2015a)), in this study we only consider the properties of anticyclones.

## 3. Results

### 3.1 Interannual variability of the Black Sea level

The variability of the Black Sea MSL is shown in Fig.1a. In 1993-2014, MSL was rising at a rate of 3.15 ±0.13 mm (black dotted line) per year, in agreement with Avsar et al. (2015). This value coincides well with the global MSL rise in 1992-2008 (e.g. Cazenave et al., 2010). The variability of the Black Sea MSL ( Fig.1a) shows that . In 1993-

2014, MSL was rising at a rate of 3.15 ±0.13 mm per year, in agreement with Avsar et al. (2015). This value coincides well with the global MSL rise in 1992-2008 (e.g. Cazenave et al., 2010). The trend has not been constant: sea level was rising in 1992-1999; then it was falling in 2000-2007; and in 2007-2014 it rose again (Fig. 1b). Similar report of sea level change during the first two periods have already been reported by Vigo et al. (2005) and Yildiz et al. (2008), based on the analysis of satellite altimetry and gravimetry data. Changes in the amount of water in the Black Sea (water balance) are the main reason for the basin-averaged sea level variability (Stanev et al., 2000, 2002; Peneva et al., 2001; Ilyin et al., 2012; Volkov and Landerer, 2015; Volkov et al., 2016). An extensive review of the Black Sea level variability and water balance in the 20th century is provided in Goryachkin and Ivanov (2006).

The spatial distribution of sea level trends in the Black Sea over the 1993-2014 time period (Fig.1c) shows that the sea level change is spatially non-uniform in agreement with earlier analyses of the along-track altimetry data (Vigo et al. 2005, Kubryakov and Stanichniy, 2013). Sea level in coastal and shelf areas was rising at rates 3.2-4±0.2 mm/year, which is approximately 1.5-2 times greater than in the center of the basin (1.5-2.5±0.25 mm/year). The largest trend exceeding 5.0±0.25 mm/year is observed in the southeastern part of the basin. The observed spatial differences in the sea level rise are related to the basin dynamics, which redistributes water mass within the basin. Investigation of the reasons of this spatial variability is the main goal of the present study.

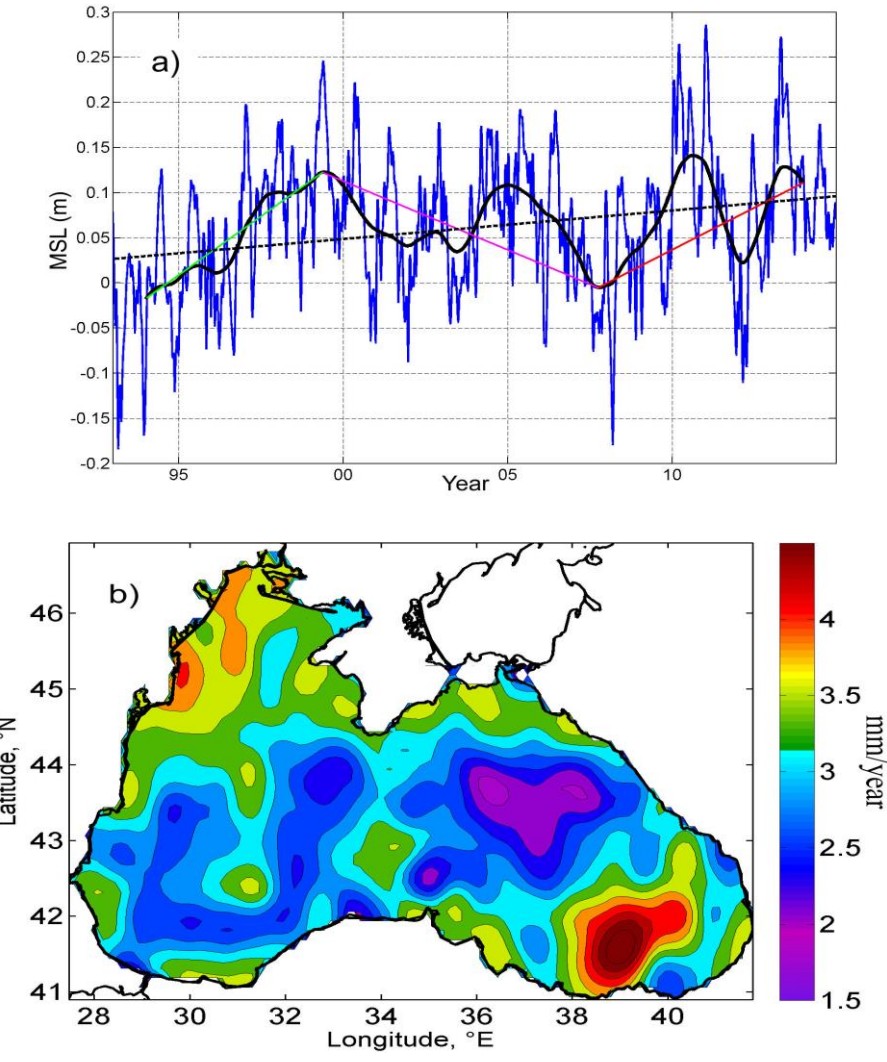

**Figure 1: a) daily time series (blue curve) of the basin-averaged Black Sea level from satellite altimetry data. Black dotted line shows the linear trend; b) spatial distribution of the 1993-2014 sea level trends (mm/year)**

**3.2 Wind-driven dynamic sea level variability**

The main feature of the Black Sea dynamics is the cyclonic Rim current encircling the basin over the continental slope. The predominantly cyclonic wind curl over the basin causes the near-surface divergence in the basin's interior and downwelling motions and associated deepening of pycnocline near the continental slope. This process generates horizontal density gradients that drive the along-slope baroclinic flow (Stanev et al., 1990, 2000; Korotaev et al., 2001).

The seasonal variability of the Black Sea DSL is driven by the seasonal changes of the wind curl (Stanev et al., 2000; Korotaev et al., 2001). In winter, the wind curl increases and intensifies the Ekman divergence, as a result the DSL falls in the center of the basin and rises at the basin periphery (Fig.2a). In summer the wind curl and divergence weakens and the water accumulated along the coast flows back into the basin's interior (Fig.2b).

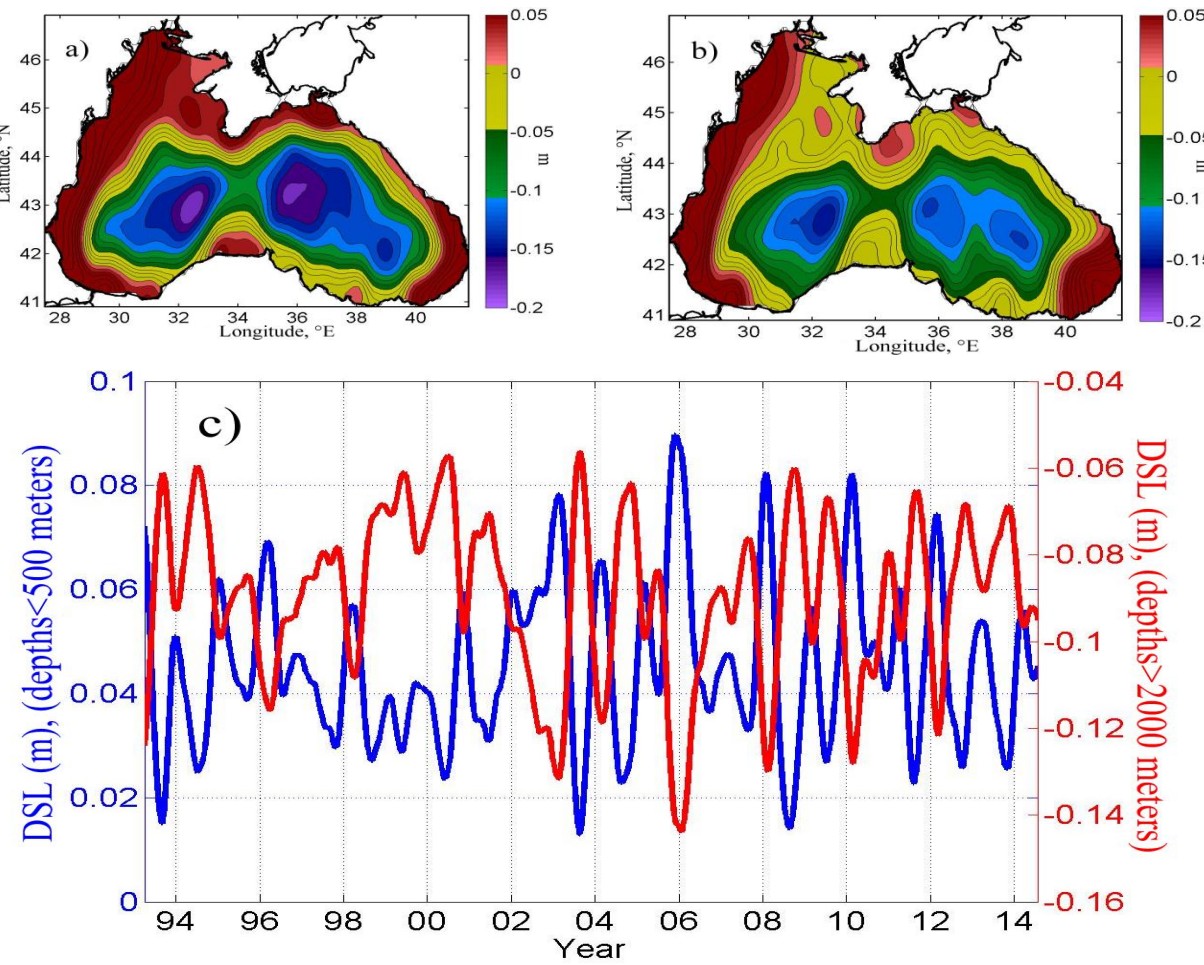

**Figure 2: Average DSL distribution in (a) February, (b) July and (c) DSL variability averaged over the central part (depths more than 2000 metres) and along the basin's periphery (depths less than 500 metres). Time series are smoothed by 90-days moving average**

By the means of Ekman dynamics, fluctuations in the wind curl over the Black Sea lead to changes in DSL also on the longer time scales: strengthening of the wind curl increase the DSL at the basin periphery and lower DSL at the basin center. As a result, the DSL in the basin's interior and periphery have an opposite variability with correlation coefficient (k=-0.91) (fig.2c) that was shown in previous studies (Stanev et al.,2000; 2001).

5       Displayed in Figure 3 is correlation map between the wind curl averaged over the basin and DSL at each grid point for the time series smoothed by a 365-day moving average (only interannual variability is retained). The correlation coefficients are significantly positive (>0.6) in shallow regions, with depths generally <500 m, and they are significantly negative in the deep interior of the basin (<-0.6). The correlation map is consistent with the second EOF of the altimetry-derived sea level, which has been attributed to the effect of the wind curl (Grayek et al.,2010).

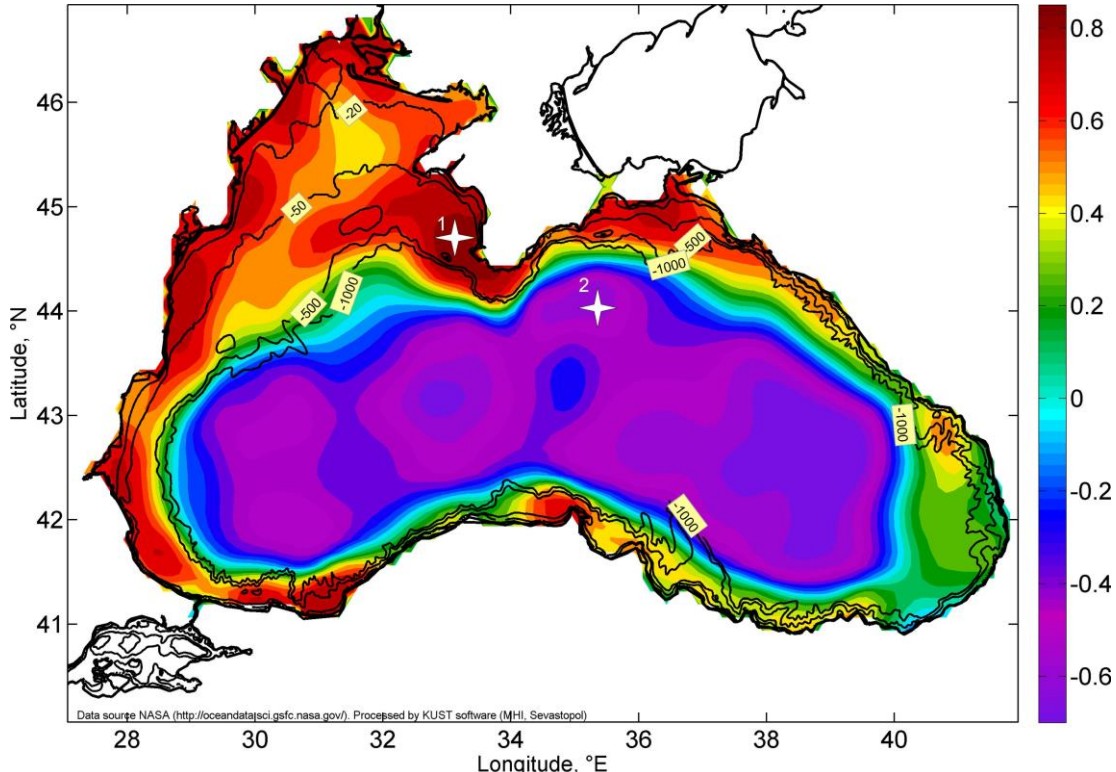

**Figure 3: Correlation coefficients between the basin-averaged wind curl and DSL for the time series smoothed with a 365-day moving average (only interannual signals are retained). Solid black lines show isobaths (20, 50, 500, 1000 metres).**

15       The correlation coefficients are high over the continental slope of the basin and the shelf areas, including the large north-western shelf area. Over the continental slope the rise of Ekman convergence  leads to the downwelling motions and lowering of the pycnocline. Related decrease of density (steric effect) and the inflow of the water from the basin center both  induce the sea level rise over the slope.  In the shallow shelf areas, where stratification is weak, at least in winter months, the observed DSL variability is primarily caused by barotropic motions. The correlation is smaller in the south-east area of the basin, which is known as the area of the Batumi eddy (Oguz et al.,

20 1993; Staneva et al., 2001; Korotaev et al., 2003, Kubryakov, Stanichny, 2015c). Here, the intense eddy dynamics can alter the large-scale DSL changes caused by Ekman transport.

       The interannual variability of the basin-averaged wind curl for the time period from 1979 to 2015 is rather complex with several sharp minima in 1983, 1990, 2000, and 2007, and several less prominent maxima (Fig.4). At the same time, it is well-seen that the wind curl is increasing over the entire period, including the period when the

25 high-accuracy altimetry measurements are available (1993-2015). The value of the linear trend is $\sim(1\pm$

$0.4) \cdot 10^{(-8)}$ 1/s per year, which constitutes about 0.5% of the average value per year. The long-term trend of the wind curl induces a long-term intensification of the basin's cyclonic circulation that is indeed observed by satellite altimetry (Kubryakov et al., 2013, 2016).. The basin-averaged speed of surface geostrophic currents $U$ (Fig.4 – blue curve) was increasing at an approximate rate of 0.05±0.003 cm/s per year, i.e. by 0.3% per year of the average value.

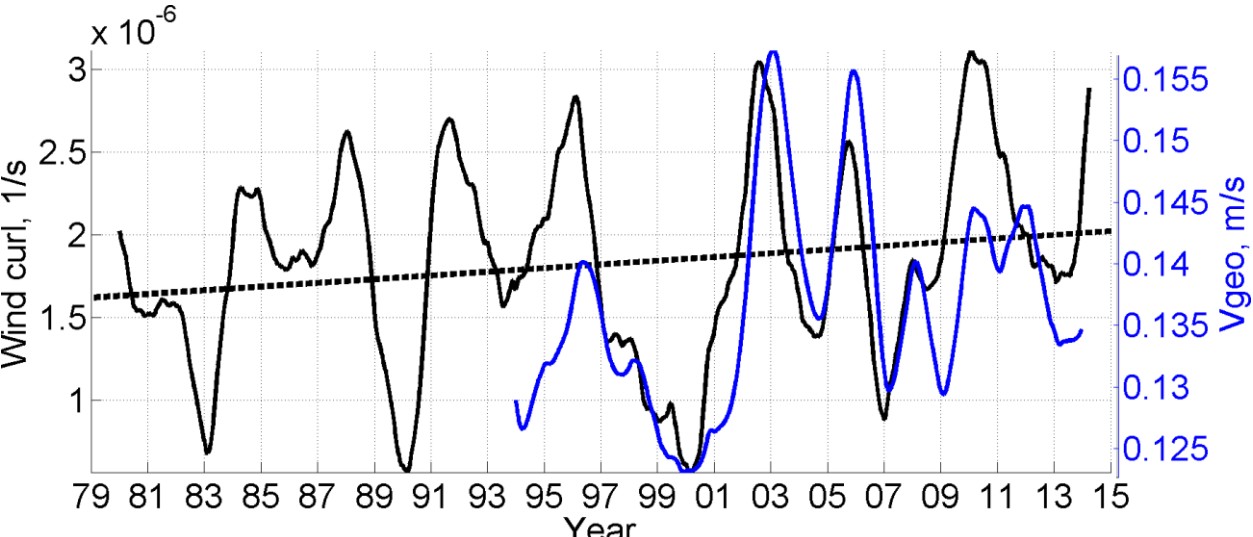

**Figure 4: The interannual variability of the basin-averaged wind curl (black curve) in 1980-2014 and the basin-averaged speed of surface geostrophic currents (blue curve) smoothed with a 1-year moving average.**

A positive trend of wind curl strengthens Ekman divergence, which lowers sea level in the center of the basin, and raises sea level along the coast. For example, figures 5 demonstrate the variability of DSL at two locations shown by crosses in Fig. 3: on the northwestern shelf (33.2°E; 44.7°N) and in the central basin (35.4°E, 44.0°N). The time series of DSL (red curve) and the basin-averaged wind curl (blue curve) are strongly correlated for the first point on both the seasonal and interannual time scales, with the time lag of about two weeks (Kubryakov et al., 2016). The correlation coefficient for the lagged time series is 0.75 for 90-day moving average smoothing, and it is 0.9 for the 365-day moving average smoothing. For the second location, characteristic for the basin's interior, the relationship between the sea level and wind curl is inverse (Fig. 5c,d). Here, the correlation is -0.84 for the time series smoothed with a 90-day moving average and -0.66 for the time series smoothed with a 365-day moving average.

The average range of the interannual oscillations of DSL at the first point is about 5 cm, in close agreement with the amplitudes of the DSL averaged along the basin's periphery (depths less than 500 metres) (Fig.2c). The seasonal amplitudes of DSL reached 10 cm in 2003 and 8 cm in 2006, 2008. Based on tide gauge measurements, the charactersitic range of seasonal oscillations of sea level at the Black Sea coastal stations is about 20 cm (Goryachkin, Ivanov, 2006). Thus, the seasonal variability of DSL explain up to 50% of the sea level variance and, therefore, makes an important contribution to the total sea level variability in agreement with previous findings (Stanev et al., 2000; Graek et al, 2010).

The linear trends of the DSL and wind curl (Fig. 5) are unidirectional (positive) in the basin's periphery and opposite in the basin's interior. The maximum increase of the cyclonic wind curl over the basin is observed in winter months (Fig. 6). As a result, the strongest intensification of the Black Sea circulation and DSL at the basin periphery (depth less than 500 metres) occurs in winter (Kubryakov et al., 2016). The time required for the Black

Sea circulation to adjust to changes in the wind curl is approximately two weeks (see Fig.5a). That is why on the graph of seasonal variability, we see that the maximum DSL trend (March) lags behind the maximum wind curl trend (February) by ~one month (Fig. 6). The similar time lag (1-2 months) between DSL and wind curl variability was obtained in an earlier study of Stanev et al. (2000). Winter-early spring months are characterized by the maximum coastal vulnerability to the DSL rise, which reaches ~1mm/year.

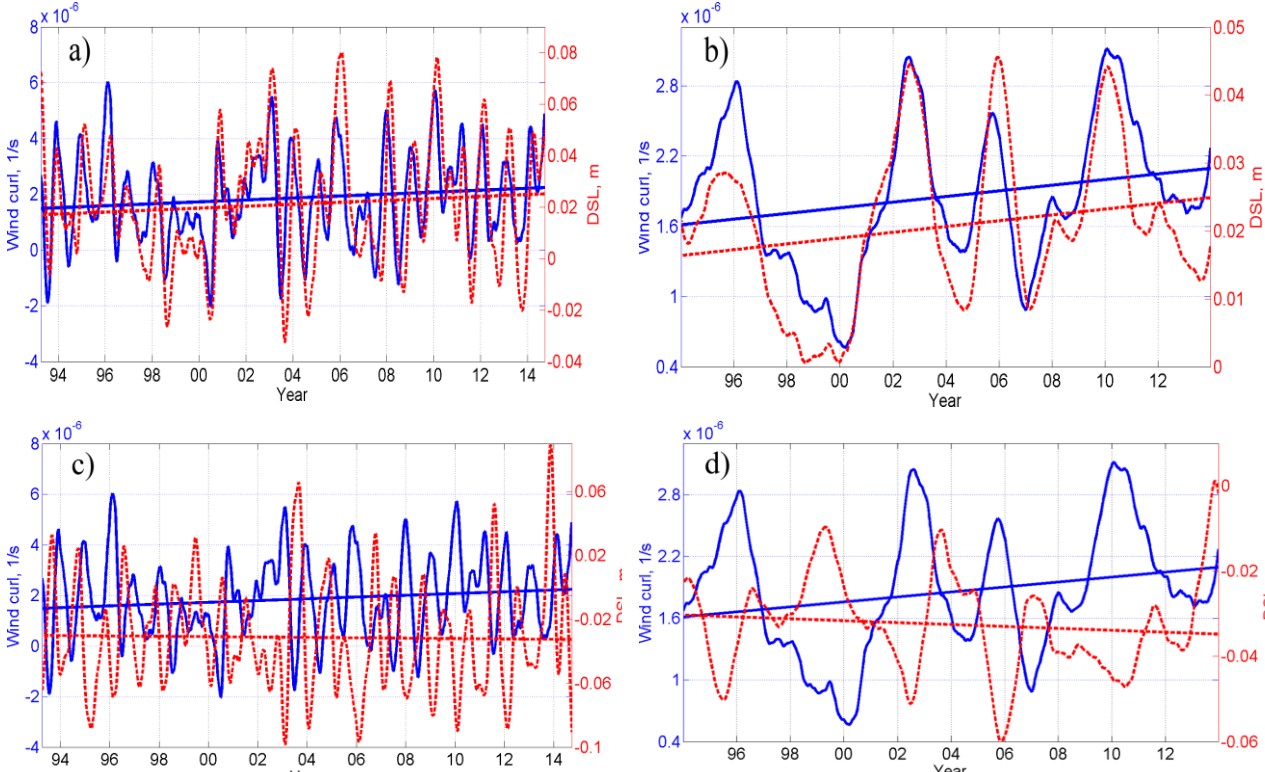

**Figure 5: The time series and the linear trend s of DSL (red curves) and the basin-averaged wind curl (blue curves): (a, b) at 33.2°E, 44.8°N  (the basin's periphery) and (c, d) at 35.4°E, 44.0°N (center of the basin); the time series are smoothed (a, c) with a 90-day moving average  time series and with a 365-day moving average.**

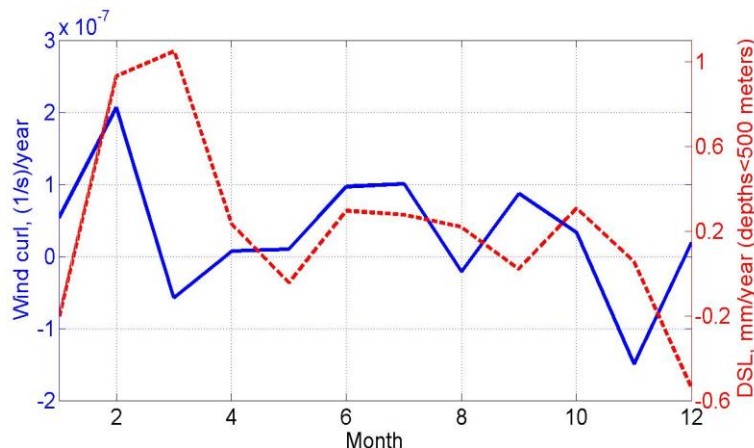

**Figure 6:  Monthly distribution of the (blue curve) wind curl (1/s per year) and (red curve) DSL trends (mm/year) at the basin periphery**

Based on the above considerations, the spatial distribution of the Black Sea trends presented in fig.1c can be explained by two factors: a) the rise of the average Black Sea level by 3.15 mm/year due to the change of water

mass/volume in the basin and b) the increase of Ekman divergence in the center of the sea due to the strengthening of the cyclonic wind curl over the basin. The magnitude of the sea level rise related to the strengthening of the large-scale circulation and the Ekman divergence in the basin can be estimated from Fig.7a, which shows the DSL=ADT-MSL trends map.   It is equal to approximately 0-0.5 mm/year at the basin periphery and approximately 1-15 mm/year in the basin's interior, The value of the DSL trend constitutes about 15%-50% of the basin-averaged sea level rise (3.15 mm/year) and, therefore, plays an important role in sea level rise estimates.

### 3.3. The impact of mesoscale variability on the sea level trends

Several localized maxima are observed in the spatial distribution of the Black Sea DSL trends (Fig. 7a The spatial pattern of the largest DSL trend (centered around 38.5°E, 42°N - Fig. 7a) is located in the southeast corner of the Black Sea in the area of quasi-stationary Batumi anticyclone(Oguz et al., 1993, Korotaev et al., 2003, Kubryakov, Stanichny, 2015c). The coincidence of the local maximum of sea level trend and the Batumi eddy position suggests that this maximum is related to the impact of eddy dynamics.

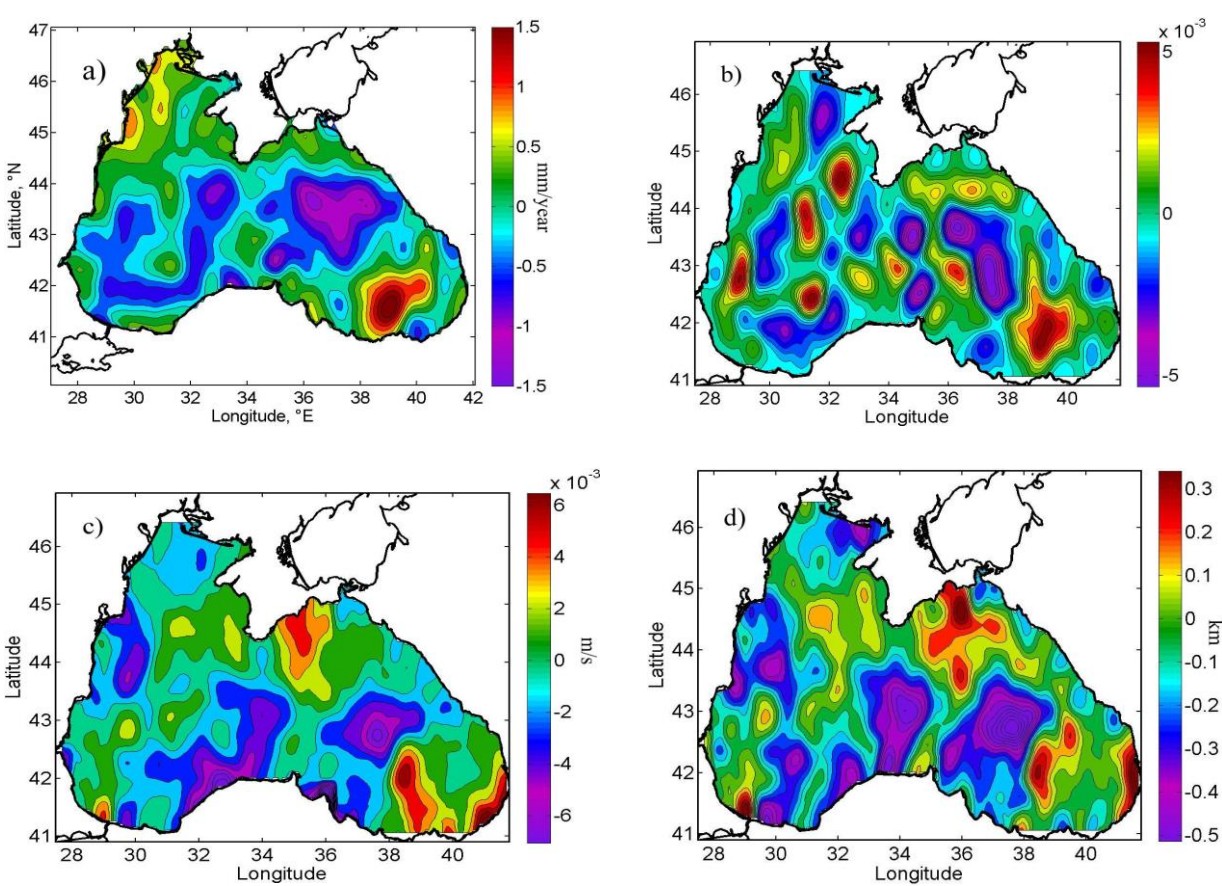

**Figure 7: Spatial distribution of a) DSL trends; b) Trends in the frequency of anticyclones in 1993-2014;  c) Trends in the radius of anticyclones in 1993-2014; d) Trends in the maximum orbital velocity of anticyclones in 1993-2014;**

Displayed in Figures 7b-7d are the linear trends of (b) frequency, (c) orbital velocity, and (d) radius of the Black sea eddies, estimated using the "winding angle" method. Although the estimated trends are rather patchy due to complex eddy dynamics, it is still possible to identify some significant trends of the radius and orbital velocity of anticyclones that coincide with the areas of stronger sea level rise. The spatial pattern of the largest eddy frequency

trend is located in the southeastern part and it coincides with the position of the risen DSL trends (Fig. 7b). Positive trends of eddy radius and orbital velocity are observed on the western and eastern sides of this pattern. This suggests that the Batumi anticyclone was expanding in the zonal direction, and began to occupy larger area in the south-east part of the basin. In the anticyclonic eddies sea level rises, as they induce convergent motions and accumulation of the surface waters in their core (see e.g. Siegel et al., 1999) . Rise and intensification of the Batumi eddy leads to the consequent rise of the dynamic sea level in the areas in which eddy expands.

The interannual variability of the anticyclonic eddy properties at the point 38.5°E, 42°N, that corresponded to the largest DSL trend in the southeast of the Black Sea is shown in Figures 8 . Both the frequency of eddies and their intensity were increasing with time. The frequency of eddies doubled from  4% in the early 1990s to ~8% in 2010s (Fig. 8a).  The maximum orbital velocity of anticyclones at this point almost tripled from 0.14 m/s in 1995 to ~0.4 m/s  in 2014 (Fig. 7b). Observed intensification of anticyclonic motions cause the largest sea level rise in the southeast corner of the Black sea. Several other local maxima in the trends of the frequency of anticyclones coincide with the positions of increased sea level trends. For example, a local DSL maximum near 31.5°E,  44°N (Fig. 7a) is close to a local maximum in the trend of the frequency and velocity of anticyclones. The variability of the eddy dynamics is one of the reasons of the observed patchiness of the Black Sea level trends.

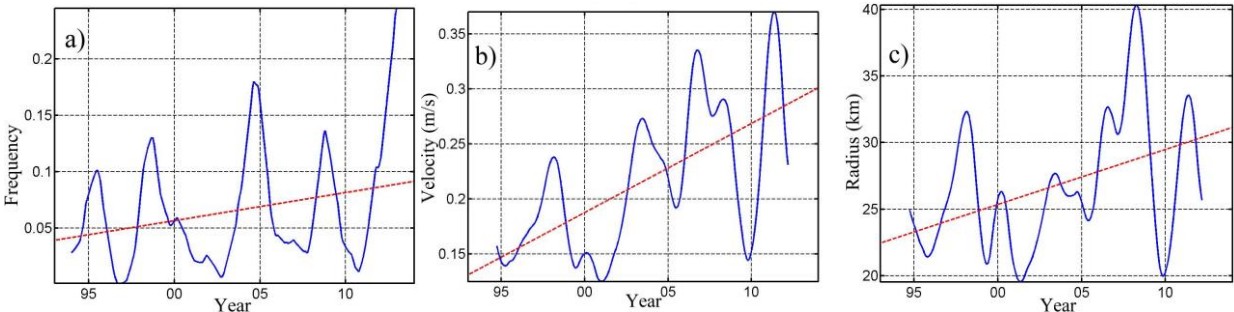

**Figure 8: a) Frequency b) the maximum orbital velocity c) radius of anticyclones at 38.5°E, 42°N, derived from altimetry data.**

### 3.4. Reconstruction of DSL variability using wind data

Before the advent of high-resolution altimetry in 1992, tide gauges were used to estimate the basin-averaged sea level rise. As demonstrated above, coastal sea level measurements include DSL, but the latter does not reflect changes in the Black Sea water volume (Stanev et al., 2001). Therefore, the tide gauge trends should be corrected for DSL in order to obtain better estimates of the basin-averaged sea level trends . To determine the DSL correction, we computed the linear regression coefficients (k) between the basin-averaged wind curl (W) and DSL at each grid point: DSL=k*W+ε  (Fig.9), where ε is the error term.

Then we reconstructed DSL using the regression coefficients and the wind curl. Standard deviations of the error term on the interannual time scale (the time series are smoothed with a 1-year moving average) are rather small along the coast and over the northwestern shelf (Fig. 9b), generally less that 1 cm. In the interior of the basin and, in particular, in the area of the Batumi anticyclone, the errors are larger (2-3 cm), which is apparently due to the impact of mesoscale dynamics. Displayed in Fig. 10 are the altimetry-derived and reconstructed DSL at a location near the south Crimean coast (33.2°E; 44.7°N). The correlation coefficients between the time series are 0.85 and 0.88 for the time series smoothed by 90-day (Fig. 10a) and 365-day (Fig. 10b) moving averages, respectively. Our analysis

suggests that a simple linear regression is capable of capturing both the seasonal and interannual variability of DSL from the wind data alone.

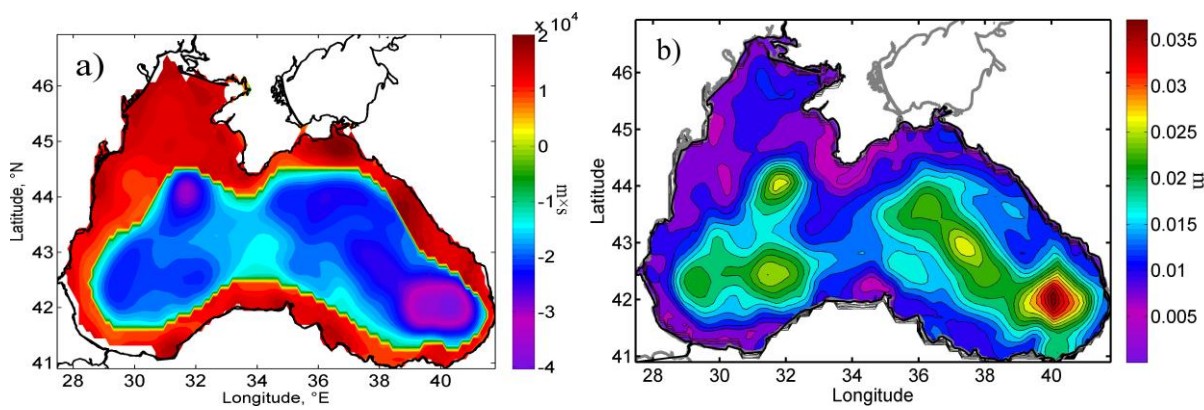

5    **Figure 9: a) Linear regression coefficients (k) between the basin-averaged wind curl and DSL (DSL=k\*W+ε); b) Standard deviations of the error term (the difference between the altimetry-derived and reconstructed DSL); time series are smoothed with a 1-year moving average.**

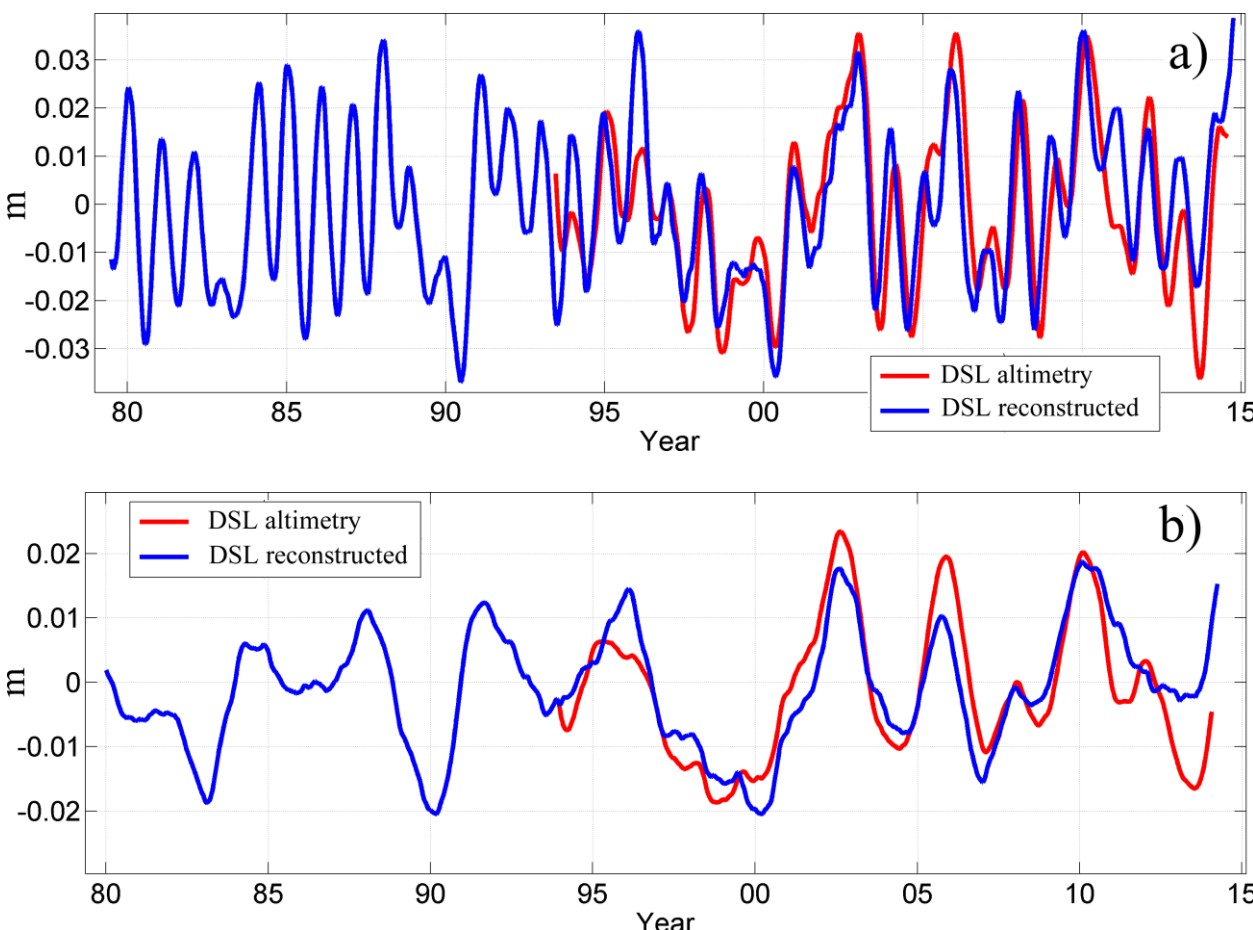

10    **Figure 10: The time series of the altimetry-derived** DSL **(red curve) and DSL** at 33.2°E, 44.7°N **reconstructed from the wind curl (blue curve): the time series are smoothed (a) with a 90-day moving average and (b) with a 365-day moving average.**

Since the ERA-Interim winds are available for a longer period (since 1979) than altimetry data, the obtained regression coefficients can be used to reconstruct the DSL variability in the past and correct the estimates of the Black Sea level rise based on tide gauges. For example, the DSL trend over the 1979-1992 period at a point near the south Crimean coast (33.2°E; 44.7°N) is 0.3 mm/year. Then, this value should be subtracted from nearby tide gauge records that are used to compute the basin-averaged sea level change in the Black Sea. It should be noted that this method accounts only for changes in the large-scale circulation, but does not account for trends in mesoscale dynamics. Nevertheless, based on our analysis it is reasonable to assume that the mesoscale dynamics mostly affects the basin's interior, while the coastal sea level variability is mostly driven by Ekman dynamics (fig. 9b).

## 4 Conclusions

The climatic changes of the large-scale and mesoscale dynamics in the Black Sea significantly impact sea level trends in different parts of the basin. While the basin-averaged sea level has been rising by 3.15 mm/year, sea level trends vary from 1.5 mm/year in the interior to 3.5-3.8 mm/year in coastal areas and to 5 mm/year in the southeastern part of the sea. We have shown that the observed long-term intensification of the cyclonic wind curl strengthened divergence in the center of the basin, which caused a rise of sea level along the Black Sea coast and over the northwestern shelf, and a lowering of sea level in the interior of the basin. In addition, we show that changes in the distribution and intensity of mesoscale eddies led to the local extremes in sea level trends. In particular, an extension of the Batumi anticyclone resulted in an excess sea level rise in the southeastern part of the basin by ~1.2 mm/year.

The DSL associated with the redistribution of water masses within the Black Sea varies considerably on seasonal and interannual time scales. For example, the maximum trend of the wind curl causing an associated DLS rise of ~1 mm/year is observed in winter months. The amplitudes of the DSL variability can reach 10 cm in different years, and they contribute up to 50% of the total annual sea level signal in agreement with Stanev et al. (2000). We have demonstrated that the DSL variability can be reconstructed using the linear regression between the wind curl and DSL. The reconstructed DSL can be used to correct historical (prior to altimetry era) estimates of the basin-averaged sea level rise, based on coastal tide gauge measurements.

**Acknowledgments**

Kubryakov A.A. was supported by RFBR, according to the research project No. 16-35-60036 mol_a_dk. Stanichny S.V. was supported by RSF, the research grant 15-17-20020. D.L. Volkov was supported by the NASA Ocean Surface Topography Science Team program (grant NNX13AO73G) and by the base funds of NOAA Atlantic Oceanographic and Meteorological Laboratory.

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
