# Peer review of "Quantifying the impact of basin dynamics on the regional sea level rise in the Black Sea"

_Ocean Science, 2016_

## Author Comment (AC1) · 7 Oct 2016

There is a mistake in D. Volkov's affiliation. The correct affiliation is 3,4: 3. Cooperative Institute for Marine and Atmospheric Studies, University of Miami, Miami FL, USA 4. NOAA Atlantic Oceanographic and Meteorological Laboratory, Miami FL, USA

---

## Referee Comment (RC1) · Anonymous Referee #1 · 23 Oct 2016

This manuscript addresses an important issue of the Black Sea oceanography. Unfortunately, the material is presented in such a way that the reader, who is not aware of the research in this field, could get an impression that the analysis of satellite altimeter data in the Black Sea and the understanding based on these data starts in 2016. I do not know whether the authors are unaware of the research in this field or they purposely presented completely unbalanced presentation of the state of the art. In both cases this leads me to the decision to reject this submission.

I admit that there is some part of novelty in this research, and if authors want to publish it, they first have to critically and in a balanced way present the advancements in the Black Sea oceanography based on the use of satellite data, as well as the advancements using satellite data in combination with tide gauge data. One example is the basic idea of the relationship between wind stress curl and sea level observed from

satellites, which is known short after the first satellite altimeter missions.. Knowing this example, I find nothing new in the statement of authors (p. 11: A simple regression allows to reconstruct both the seasonal and interannual variability of DSL from the wind data alone.) The second example is the dynamics of coastal and open-ocean sea level (Fig. 2d).

Authors have to make a full inventory of what has been done in this field, clearly identify what is known from the past studies, what has not been solved, or has been addressed wrongly, and finally they have to clearly describe their step forward. For me there is almost nothing new in the first part of their manuscript, just a repetition of older research based on longer in time records. I would ask what new they would find when using about a 20-year long data set.

Authors say: "The strong coincidence between basin bathymetry and correlations patterns is well seen". They have to know that this is not coincidence at all. Fig. 3 is just an illustration of the role of the Ekman pumping, which is largely addressed in the Black Sea literature. This "coincidence" reflects the dynamics of pycnocline (sea level just mirrors it).

---

## Referee Comment (RC2) · Anonymous Referee #2 · 27 Oct 2016

This paper investigates changes in the Black Sea circulation as evidenced by altimetry. These changes are linked, as expected, with the wind forcing which is then used to reconstruct this variability for a period before altimetry started. The work is interesting but not presented carefully and detailed enough and includes a number of significant omissions and misinterpretations. Therefore it cannot be published in its present form. It will require rethinking and rewriting so major revision is recommended. But there is merit in it and can become a useful addition to the existing literature after careful consideration.

Suggested changes: 1. The title would have been better if it was something like "Inter-annual and decadal changes in the circulation of Black Sea as evidenced from altimetry". The suggested sea level trends are neither basin wide trends nor coastal trends. 2. lines 8-10: altimetry does not measure at coastal areas. Either tide-gauges should

be used to substantiate a difference between coastal and open-sea sea level variance or this statement should be changed. 3. Lines 11-14: If the explanation concerns the period 1993-2014 the relevant forcing should be the same not a different time period. 4. Lines 16-19: How do you know that the variability is "well reconstructed" for the period before altimetry as you have no data? 5. Lines 18-19: Why do tide gauges need corrections for what happens away from the coast? They provide direct measurements of sea level. In any case as altimetry does not provide information closer to ∼30km from the coast this suggestion is erroneous. 6. There is significant literature concerning sea level rise for the Black Sea (for example Stanev et al., 2000; 2002; Tsimplis et al, 2004 and Volkov and Landerer- which is referenced ) discuss sea level rise in the Black Sea and assessing mass addition to the basin as well as steric effects. These are more relevant than a general discussion of what causes global sea level rise. 7. Section 2 data. Need to describe the dataset properly. While there is a paper (Volkov and Landerer, 2015) which argues that the altimetry data set can be used as is in the Black Sea with the imposed DAC for pressure and wind, their argument is based on comparison with tide gauges and their finding that such a correction does not improve the agreement with tide gauges in RMS terms of monthly values. This does not necessarily mean that there are no "trends" is the pressure and wind fields which are artificially and in a spatially coherent manner added as a correction to the se level field through DAC. Thus in, my view, the physical argument that the constraints imposed by the Turkish Straits to water exchange do not permit the use of DAC is the correct one. The argument about RMS change can only partly justify the use of correction and probably not in the context of trends. In addition to the doubts I have in relation to the atmospheric correction it is unclear which other corrections are used and what is their uncertainty. Do the data have a GIA correction and how large it is? While it is not likely to be large it will provide confidence to the data process to express it clearly. 8. The general uncertainties on the altimetry trends need also to be addressed. While the uncertainty for global trends has been stated to be 0.4-0.6 mm/yr (with one exception of 0.9 mm/yr) several statements about larger uncertainties in regional trends exist. An

uncertainty of 1 mm/yr would render some of the suggested spatial variance in trends insignificant though of course there are some strong gradients demonstrated. 9. The same point about uncertainty and trends holds for all the physical parameters used. Trends are stated without much consideration of their significance. 10. My understanding of the circulation features of the black Sea suggests strong seasonality. This paper does not deal with this at all. Are these trends consistent during the year or are they an expression of strengthening of seasonal circulation? This requires extra work. 11. The figures should demonstrate the limitations of altimetry by leaving the 30-40 km near the coast blank rather than closing the contouring. This is done for figures 6b,c and d but not for Fig 6a or any other contour plot. With the Black Sea at around 260km at its narrowest having 60-80 km of information lost is a significant percentage of area. 12. The straight lines at Figure, 5 and 7b (trends) are not persuasive. A step change seems also a good alternative.

———————————————

---

## Author Response (AR1)

**Answer to Reviewer 1**

**Answer:** Thank You for your comments. At first we would like to clarify - the main goal of the study is quantitative estimation of spatial heterogeneity of sea level rise (i.e. average trends of sea level – important climatic signal) in the Black Sea and its relation with dynamic processes.

We don't discuss the water balance, we do not want to introduce the new mechanism driving the Black Sea dynamics, because it is well known from previous studies. We provide this information in the article text during the discussion of the "3.2 Dynamic sea level variability": Here we provide the references on the major researches (not all, of course) dedicated to the study of the impact of the wind curl on the dynamic sea level. Sorry, during the manuscript preparation, we missed one really important reference "Stanev, E. V., P.-Y. Le Traon, and E. L. Peneva, 2000, Seasonal and interannual variations of sea level and their dependency on meteorological and hydrological forcing. Analysis of altimeter and surface data for the Black Sea, J. Geoph. Res., 105, 17203-1721", which is one of the basic studies of altimetric sea level in the Black Sea . However, the results of the (Stanev et al., 2000) are advanced in the later study (Graek et al., 2010) that is cited in the text.

All cited studies provide an explanation about the simple mechanism of the reaction of the basin sea level on the change of the cyclonic wind curl: wind curl intensify the cyclonic circulation, as a result sea level rises on periphery and decreases in the basin center. Again, this is well known issue of the Black sea dynamics, and we do not want to "discover" it.

However, we feel that it is important to illustrate this mechanism to the reader. That is why we provide the figure 2, that brings no new information, but just needed for illustrative purposes (for our opinion). This is in agreement with reviewer comments.

The new in this article is the description of the long-term trends of the Black Sea level related to the long-term trends of the Black Sea dynamics and wind changes. We, for the first gave the quantitive estimations of the impact of the large-scale and mesoscale circulation changes (DSL) on the Black sea level rise and its spatial heterogenity. This is the main novelty of the manuscript.

*According to Your comments in the revised version of the manuscript we change the Introduction part completely to more clearly define the manuscript goals and the state of art in this field. We also significantly extend the reference list and we noticeable change the part of the manuscript dedicated to the impact of the wind curl on the large-scale dynamic sea level variability.*

Below we provide answers on the reviewer comments step by step:

**1) Reviewer:** "Unfortunately, the material is presented in such a way that the reader, who is not aware of the research in this field, could get an impression that the analysis of satellite altimeter data in the Black Sea and the understanding based on these data starts in 2016. "

**Answer:** This is not exactly true, as it was shown above. Article has at least 20 references on papers interpreted altimetry data for the Black Sea since 2001 till 2016. Particularly, one of the first paper describing the Black Sea level from altimetry data is Korotaev, 2001, that is cited in the text. During the discussion of the Black Sea dynamic sea level variability we missed one really important reference "Stanev, E. V., P.-Y. Le Traon, and E. L. Peneva, 2000, Seasonal and interannual variations of sea level and their dependency on meteorological and hydrological forcing. Analysis of altimeter and surface data for the Black Sea, J. Geoph. Res., 105, 17203-1721", which is one of the basic studies on altimetric sea level in the Black Sea. However, the results of the (Stanev et al., 2000) are advanced in the later study (Graek et al., 2010) that is cited in the text.

As, the main goal of this study is the investigation of the sea level trends, the introduction is mostly dedicated to the studies of the Black sea level rise. The review of the studies dedicated to the dynamic sea level variability in the Black Sea is given in section "3.2 Dynamic sea level variability". We believe moved this part in the Introduction to avoid the false impression.

*According to Your comments we change the Introduction part completely to more clearly define the manuscript goals and the state of art in this field. We also significantly extend the reference list and we noticeable change the part of the manuscript dedicated to the impact of the wind curl on the large-scale dynamic sea level variability.*

**2) Reviewer:** "I do not know whether the authors are unaware of the research in this field or they purposely presented completely unbalanced presentation of the state of the art… One example is the basic idea of the relationship between wind stress curl and sea level observed from satellites, which is known short after the first satellite altimeter missions…. The second example is the dynamics of coastal and open-ocean sea level (Fig. 2d)."

**Answer:** Also, we cannot fully agree with the reviewer. The information about previous researches on the dynamic sea level variability is given in section "3.2 Dynamic sea level variability". We provide a shot review about the previous and modern studies, which highlight the basic idea of the relationship between wind stress curl and sea level in a first paragraph:

"The main feature of the Black Sea dynamics is the cyclonic Rim current encircling the basin over the continental slope. As a result of the cyclonic circulation, the DSL is lower in the center

of the basin and higher along the periphery (Oguz et al., 1993; Korotaev et al., 2001). The seasonal variability of the Black Sea circulation is driven by changes in the wind curl averaged over the basin. (Stanev, 1990; Korotaev, 2001, Graek et al., 2010). In winter, the cyclonic wind curl and, therefore, the onshore Ekman transport increase and cause divergence in the center of the basin by moving water to the basin's periphery The compensating vertical uplift (Ekman suction) in the center of the sea brings dense deep water masses to the surface, while light surface waters move towards the coast (Korotaev, 2001; Kubryakov et al., 2016). The redistribution of mass and volume results in a decrease of sea level in the basin's center, and an increase along the coastline. In summer, the cyclonic wind curl weakens, Ekman divergence decreases and the water accumulated along the coast flows back into the basin's interior (Zatsepin et al., 2002; Kubryakova, Korotaev, 2016)."

The figure 2 is given to illustrate the basic ideas given in the cited studies, which are crucial to understand the impact of the wind curl on the sea level rise. For our opinion, it is useful for the illustrative purposes. Some readers can be unfamiliar to the Black Sea dynamics (for example specialists that work with tide gauges data), that is why figure 2 is in the text.

We believe that we can improve the phrase in the text:" As expected, the seasonal time series of DSL in the coastal (depths less than 500 meters) and central (depths more than 2000 meters) parts of the basin are negatively correlated (fig.2d). " to "As it is known (Stanev et al., 2000; Korotaev et al., 2001), the seasonal time series of DSL in the coastal (depths less than 500 meters) and central (depths more than 2000 meters) parts of the basin are negatively correlated (fig.2d)."

*We significantly extend the reference list and we noticeable change the part of the manuscript dedicated to the impact of the wind curl on the large-scale dynamic sea level variability.We replace the phrase :" As expected, the seasonal time series of DSL in the coastal (depths less than 500 meters) and central (depths more than 2000 meters) parts of the basin are negatively correlated (fig.2d). " to "By the means of Ekman dynamics, fluctuations in the wind curl over the Black Sea lead to changes in DSL also on the longer time scales: strengthening of the wind curl increase the DSL at the basin periphery and lower DSL at the basin center. As a result, the DSL in the basin's interior and periphery have an opposite variability with correlation coefficient (k=-0.91) (fig.2c) that was shown in previous studies (Stanev et al.,2000; 2001)."*

**3) Reviewer:** Knowing this example, I find nothing new in the statement of authors (p. 11: A simple regression allows to reconstruct both the seasonal and interannual variability of DSL from the wind data alone.)

**Answer:** As far as we know, the reconstruction of the DSL spatial field from the wind curl data on the interannual time scales was not done before. We'll be very appreciated to obtain references on previous study demonstrated such reconstruction. We think that the results obtained are important for the correction of the sea level rise from the historical tide gauges measurements.

**4) Reviewer::** "The strong coincidence between basin bathymetry and correlations patterns is well seen". They have to know that this is not coincidence at all. Fig. 3 is just an illustration of the role of the Ekman pumping, which is largely addressed in the Black Sea literature. This "coincidence" reflects the dynamics of pycnocline (sea level just mirrors it)."

**Answer:** We agree that "Coincidence" is not good term, "similar spatial patterns" is better. However the reviewer not exactly true interprets observed phenomena. The dynamics of pycnocline is the secondary process defined by wind curl. The wind curl determines the intensity of water divergence from the center to the periphery. The redistribution of the sea level driving by the Ekman transport causes the downwelling motions over the continental slope, and consequent pycnocline displacement.

Moreover, in the shelf areas, (for example in the very large North-Western shelf), where the correlation is also positive and high, we can not talk about pycnocline at all. There is no main pycnocline in this shallow zone. In winter there is no stratification at all, and the dynamic sea level redistribution is caused by barotropic motions.

That is why in manuscript we wrote:

"The seasonal variability of the Black Sea circulation is driven by changes in the wind curl averaged over the basin (Stanev, 1990; Korotaev, 2001, Graek et al., 2010). In winter, the cyclonic wind curl and, therefore, the onshore Ekman transport increase and cause divergence in the center of the basin by moving water to the basin's periphery. The compensating vertical uplift (Ekman suction) in the center of the sea brings dense deep water masses to the surface, while light surface waters move towards the coast (Korotaev, 2001; Kubryakov et al., 2016). The redistribution of mass and volume results in a decrease of sea level in the basin's center, and an increase along the coastline. In summer, the cyclonic wind curl weakens, Ekman divergence decreases and the water accumulated along the coast flows back into the basin's interior (Zatsepin et al., 2002; Kubryakova, Korotaev, 2016)."

And again "Therefore, the spatial distribution of the Black Sea trends presented in fig.1c can be explained by two factors: a) the rise of the mean Black Sea level by 3.15 mm/year due to the change of water mass/volume in the basin and b) the increase of Ekman divergence in the center of the sea due to the strengthening of the cyclonic wind curl over the basin"

*In the revised version of the manuscript we rewrite this part of a text as:"The correlation coefficients are high over the continental slope of the basin and the shelf areas, including large north-west shelf. Over the continental slope the rise of DSL leads to the downwelling motions, lowering of the pycnocline, that drives the Rim current. In the shallow shelf areas, where stratification is weak, at least, in winter months, the observed DSL variability is primarily caused by barotropic motions. "*

**5) Reviewer:** I would ask what new they would find when using about a 20-year long data set.

**Answer:** The main new results of the paper are:

1)      The impact of the long-term wind curl change on the intensification of the Black Sea large-scale circulation and increase of the sea level rise in the coastal zone

2)      The impact of the changes of the Black sea mesoscale circulation on the sea level rise in the basin

3)      The reconstruction of the DSL spatial fields using reanalysis data. It should be used to correct previous estimates of the Black sea level rise in the basin

4)      Spatial distribution of the Black Sea dynamic sea level trends

**Answer to Reviewer 2**

This paper investigates changes in the Black Sea circulation as evidenced by altimetry. These changes are linked, as expected, with the wind forcing which is then used to reconstruct this variability for a period before altimetry started. The work is interesting but not presented carefully and detailed enough and includes a number of significant omissions and misinterpretations. Therefore it cannot be published in its present form. It will require rethinking and rewriting so major revision is recommended. But there is merit in it and can become a useful addition to the existing literature after careful consideration.

Suggested changes:

1. The title would have been better if it was something like "Interannual and decadal changes in the circulation of Black Sea as evidenced from altimetry". The suggested sea level trends are neither basin wide trends nor coastal trends.

**Answer:** The goal of this study is to investigate the sea level trends in the basin with the focus on its spatial heterogeneity. It is not dedicated to the study of the Black sea dynamics, which was investigated earlier in the number of studies (e.g. Stanev et al., 2000, 2001; Korotaev, 2001; 2003, Kubryakov et al., 2016.) In this study we provide the quantitative estimates of the spatial variability of the sea level rise in the basin and describe the main reasons of its heterogeneity, which are the long-term changes of the Black sea large-scale and mesoscale dynamics. This is the main novelty of the study, which is dedicated to the understanding of the effects of the Black sea dynamics on the sea level rise in the basin.

*Due to Your comments we decided to change the manuscript title on "Impact of basin dynamics on the regional sea level trends in the Black Sea"*

2. lines 8-10: altimetry does not measure at coastal areas. Either tide-gauges should be used to substantiate a difference between coastal and open-sea sea level variance or this statement should be changed.

**Answer:** We respectfully disagree with the reviewer. Altimetry does measure near the coast, but these measurements are less accurate being constrained by the size of the altimeter footprint. Nevertheless, in the recent years, a great progress in improving the near-coast measurements has been achieved, which has affected the regional altimetry products, such as the Mediterranean and Black Sea products. The improvement in the coastal areas of the Mediterranean Sea has recently been demonstrated by Marcos et al. (Advances in Space Res., 2015). The nearest points of the altimetry along-track measurements is situated at ~ 7

km distance from the coast (see fig.S1 in the attached file). The resolution of the Black sea mapped regional product is 1/8° or ~12.5 km. Regional Black Sea array of mapped altimetry sea level anomalies (MSLA) is produced by the CLS Space Oceanography Division and distributed by Aviso, with support from Cnes (http://www.aviso. oceanobs.com/). In order to be precise we should change the phrase in the Introduction "the sea level rise varied from 0.15-2.5 mm/year in the central part to 3.5-3.8 mm/year in coastal areas and 5 mm/year in the southwestern part of the sea" to "the sea level rise varied from 0.15-2.5 mm/year in the central part to 3.5-3.8 mm/year at the periphery of the basin and 5 mm/year in the southwestern part of the sea"

[Figure]

Fig.1 Left - Track position of ERS-2 altimetric measurements from the Black Sea regional dataset; Right – track position of the Saral\Altika near the Crimea. The distance between coast and nearest coastal point is ~ 7 km

*We significantly extend the description of the altimetry dataset in the revised version of the manuscript and add information about near-coast measurements of modern altimeters*

3. Lines 11-14: If the explanation concerns the period 1993-2014 the relevant forcing should be the same not a different time period.

**Answer:** We agree with the reviewer. The phrase "A long-term increase of the cyclonic wind curl over the basin from 1979 to 2014 strengthened divergence in the center of the Black Sea that led to an increase of sea level near the coast and a decrease in the center of the basin" should be changed to "A long-term increase of the cyclonic wind curl over the basin strengthened divergence in the center of the Black Sea that led to an increase of sea level near the coast and a decrease in the center of the basin"

*We changed the abstract according to Your comments*

4. Lines 16-19: How do you know that the variability is "well reconstructed" for the period before altimetry as you have no data?

**Answer:** We agree with the reviewer. The phrase "The DSL variability in the Black Sea depends strongly on the basin-averaged wind curl and is well reconstructed using the ERA-Interim winds from 1979 to present, including the time when altimetry data was unavailable. The reconstruction can be used to correct historical tide gauges data for dynamic effects, which are usually neglected in the analysis of the Black Sea tide gauge records." should be changed to "The DSL variability in the Black Sea depends strongly on the basin-averaged wind curl. In the study we show that the DSL variability on interannual and seasonal time scales can be reconstructed with a reasonable accuracy using simple linear regression of wind curl data from atmospheric reanalysis. Before the emergence of altimetry data the measurements at the periphery of the basin (e.g. coastal tide gauges) were used to estimate the basin-averaged sea level rise. As the DSL trends at the basin periphery do not reflect the change of mean water volume, they should be subtracted for the correct estimation of the basin-averaged sea level trends. The method presented in the study can be used to correct historical estimates of basinaveraged sea level rise on dynamic effects using atmospheric reanalysis data."

*We changed the abstract according to Your comments*

5. Lines 18-19: Why do tide gauges need corrections for what happens away from the coast? They provide direct measurements of sea level. In any case as altimetry does not provide information closer to ~30km from the coast this suggestion is erroneous.

**Answer:** We agree with the reviewer. This statement should be rewritten more precisely. The estimates of the basin-averaged sea level rise from tide gauges needs correction on the dynamic effects, not tide gauges themselves. DSL trends impact on the estimates of the sea level rise, if we measure only at the periphery of the basin. This impact can be subtracted using given in the study method. The nearest points of the altimetry along-track measurements is situated at ~ 7 km distance from the coast (see fig.S1 in the attached file). The resolution of the Black sea mapped regional product is 1/4° or ~12.5 km. Several studies have shown that the data in the closest point of altimetry-track is well correlated with the tide gauges measurements (Korotaev et al., 1998 (in russian); Stanev et al., 2000; Peneva et al., 2001; Goryachkin et al., 2001, 2003 (in russian); Kubryakov et al., 2013; Avsar et al., 2015; Volkov and Landerer, 2015).

*We changed the abstract according to Your comments and add information about near-coast measurements of modern altimeters in the Section 2*

6. There is significant literature concerning sea level rise for the Black Sea (for example Stanev et al., 2000; 2002; Tsimplis et al, 2004 and Volkov and Landerer- which is referenced ) discuss sea level rise in the Black Sea and assessing mass addition to the basin as well as steric effects. These are more relevant than a general discussion of what causes global sea level rise.

**Answer:** We agree with the reviewer. Several references should be added to the introduction to better represent the previous research.

*According to Your comments we change the Introduction part completely and significantly extend the reference list.*

7. Section 2 data. Need to describe the dataset properly. While there is a paper (Volkov and Landerer, 2015) which argues that the altimetry data set can be used as is in the Black Sea with the imposed DAC for pressure and wind, their argument is based on comparison with tide gauges and their finding that such a correction does not improve the agreement with tide gauges in RMS terms of monthly values. This does not necessarily mean that there are no "trends" is the pressure and wind fields which are artificially and in a spatially coherent manner added as a correction to the se level field through DAC. Thus in, my view, the physical argument that the constraints imposed by the Turkish Straits to water exchange do not permit the use of DAC is the correct one. The argument about RMS change can only partly justify the use of correction and probably not in the context of trends. In addition to the doubts I have in relation to the atmospheric correction it is unclear which other corrections are used and what is their uncertainty. Do the data have a GIA correction and how large it is? While it is not likely to be large it will provide confidence to the data process to express it clearly.

**Answer:** We should point out that the main purpose of applying the DAC correction to altimetry data is to reduce the aliasing that results from the barotropic response of the ocean to the variable high frequency atmospheric forcing. For this reason, the use of the DAC correction is necessary. The DAC combines the high frequency bands (periods 20 days) from the inverted barometer (IB) correction. We agree with the reviewer that changes in sea level pressure over the Black Sea (including trends) enter the DAC IB correction, and this can introduce spurious sea level changes. However, this is exactly what Volkov and Landerer (2015) addressed by adding the IB correction back to altimetry data and comparing these

data to tide gauges. The result shows that adding back the IB correction does not significantly improve the comparison. The RMS differences reported in the paper refer to month-to-month changes as well as to trends. Note that according to a recent study by Volkov et al (2016 – referenced in the manuscript), the constraints imposed by the Turkish Straits are not anymore effective at the interannual and longer time scales, at which the Black Sea level responds to sea level pressure changes in a pure inverted barometer manner. The use of GIA correction would be necessary for tide gauge records, but we did not use tide gauges in our paper. Please note that we use a standard altimetry product that is routinely corrected for instrumental errors and geophysical effects. It is beyond the scope of our manuscript and not necessary for the objectives of the study to present details on the corrections applied to the altimetry product (the details can be found in dedicated literature that is referenced on the AVISO web site).

*In the revised version of the manuscript we significantly extend the description of the altimetry dataset, add information about altimetry data processing, near-coast measurements of modern altimeters and its comparison with tide gauges data in the Black Sea.*

8. The general uncertainties on the altimetry trends need also to be addressed. While the uncertainty for global trends has been stated to be 0.4-0.6 mm/yr (with one exception of 0.9 mm/yr) several statements about larger uncertainties in regional trends exist. An uncertainty of 1 mm/yr would render some of the suggested spatial variance in trends insignificant though of course there are some strong gradients demonstrated.

**Answer:** We agree with the reviewer. We will add the estimates of the data uncertainties and trend uncertainties to the revised version of the paper.

*We add the estimates of the trend uncertainties in the revised version of the manuscript*

9. The same point about uncertainty and trends holds for all the physical parameters used. Trends are stated without much consideration of their significance.

**Answer:** We agree with the reviewer. We will add the estimates of the trend uncertainties to the revised version of the paper.

*We add the estimates of the trend uncertainties in the revised version of the manuscript*

10. My understanding of the circulation features of the Black Sea suggests strong seasonality. This paper does not deal with this at all. Are these trends consistent during the year or are they an expression of strengthening of seasonal circulation? This requires extra work.

**Answer:** The Section 3.2 (Paragraph 1-3) and figure 2 describes the seasonal variability of the Black Sea dynamic sea level. The increase of the cyclonic wind curl on interannual time scale leads to the intensification of the basin cyclonic circulation, as a result sea level rises on periphery and decreases in the basin center. This effect is well seen on the interannual time scales for the time series smoothed with a 365-day moving average (see fig.4 b,d; fig.5 b,c), i.e. this effect is observed for yearly-averaged data. The seasonal variability does not affect the estimates of the average DSL trends.

*Due to your comments we decided to add the analysis of the seasonal changes of the DSL trends in the manuscript. These analysis shows that the winter-early spring months are characterized by the maximum coastal vulnerability to the DSL rise, which reaches ~1mm/year. Thank You, we believe it is a valuable result for our studies.*

11. The figures should demonstrate the limitations of altimetry by leaving the 30-40 km near the coast blank rather than closing the contouring. This is done for figures 6b,c and d but not for Fig 6a or any other contour plot. With the Black Sea at around 260km at its narrowest having 60-80 km of information lost is a significant percentage of area.

**Answer:** As we mentioned above, altimetry provides measurements near the coast and the accuracy of these measurements has been improved. A reasonable comparison with tide gauges (Volkov and Landerer, 2015) suggests that the use of near-shore data points in AVISO product is justified. Therefore, we respectfully disagree with the reviewer and decide not to leave the nearshore regions blank. The nearest points of the altimetry along-track measurements is situated at ~ 7 km distance from the coast (see fig.S1). Several studies have shown that the data in the closest point of altimetrytrack is well correlated with the tide gauges measurements (Korotaev et al., 1998 (in Russian); Stanev et al., 2000; Peneva et al., 2001; Goryachkin et al., 2001, 2003 (in russian); Kubryakov et al., 2013; Avsar et al., 2015; Volkov and Landerer, 2015). In this study we use the standard mapped satellite sea level anomaly product without any extrapolation. The resolution of the Black sea mapped regional product is 1/8° or ~12.5 km.

*In the revised version of the manuscript we add some information about altimetry data processing and its comparison with tide gauges data*

12. The straight lines at Figure, 5 and 7b (trends) are not persuasive. A step change seems also a good alternative.

**Answer:** We agree that in these figures the linear trends can vary for different periods of time. However, the main task of this study is to understand the spatial variability of the Black sea level trends during the whole investigation period. That is why in figures 5 and 7b we use approximation by linear function to understand the average changes of the investigated parameters.

[revised manuscript text omitted]

Le Traon P-Y., F. Nadal, N. Ducet (1998), An improved mapping method of multisatellite altimeter data. J. Atmos. Oceanic Technol. 15:522–534. doi:10.1175/1520-0426(1998).

Lynch, D. R., & Gray, W. G. (1979). A wave equation model for finite element tidal computations. Computers & fluids, 7(3), 207-228.

Marcos, M., Pascual, A., & Pujol, I. (2015). Improved satellite altimeter mapped sea level anomalies in the Mediterranean Sea: A comparison with tide gauges.Advances in Space Research, 56(4), 596-604.

Oguz, T., V. S. Latun, M. A. Latif, V. V. Vladimirov, H. I. Sur, A. A. Makarov, E. Ozsoy, B. B. Kotovshchikov, V. Eremeev, and U. Unluata (1993), Circulation in the surface and intermediate layers of the Black Sea, Deep Sea Res.,Part I, 40, 1597– 1612.

Peneva, E., Stanev, E., Belokopytov, V., & Le Traon, P. Y. (2001). Water transport in the Bosphorus Straits estimated from hydro-meteorological and altimeter data: seasonal to decadal variability. Journal of Marine Systems,31(1), 21-33.

Palanisamy, H., Cazenave, A., Delcroix, T., & Meyssignac, B. (2015). Spatial trend patterns in the Pacific Ocean sea level during the altimetry era: the contribution of thermocline depth change and internal climate variability. Ocean Dynamics, 65(3), 341-356.

Prandi P., Cazenave A. and Becker M. Is coastal mean sea level rising faster than the global mean? A comparison between tide gauges and satellite altimetry over 1993-2007 / Geophysical Research Letters. – 2009. – V.36. - pp. 5602–5606

Reva, Y. A. (1997). Interannual oscillations of the Black Sea level. Oceanology of the Russian Academy of Sciences, 37(2), 193-200.

Stanev EV (1990) On the mechanisms of the Black Sea circulation. Earth-Sci Rev 28:285–319

Stanev, E. V., Le Traon, P. Y., & Peneva, E. L. (2000). Sea level variations and their dependency on meteorological and hydrological forcing: Analysis of altimeter and surface data for the Black Sea. Journal Of Geophysical Research-Oceans, 105(C7), 17203-17216.

Stanev, E. V., & Peneva, E. L. (2001). Regional sea level response to global climatic change: Black Sea examples. Global and Planetary Change, 32(1), 33-47.

Tsimplis, M. N., & Spencer, N. E. (1997). Collection and analysis of monthly mean sea level data in the Mediterranean and the Black Sea. Journal of Coastal Research, 534-544.

Tsimplis, M. N., Josey, S. A., Rixen, M., & Stanev, E. V. (2004). On the forcing of sea level in the Black Sea. Journal of Geophysical Research: Oceans, 109(C8).

Vigo, I., Garcia, D., Chao, B.F. Change of sea level trend in the Mediterranean and Black seas / Journal of Marine Research. – 2005. – V. 63 - pp. 1085-1100

Volkov, D. L., Larnicol, G., & Dorandeu, J. (2007). Improving the quality of satellite altimetry data over continental shelves. Journal of Geophysical Research: Oceans, 112(C6).

Volkov, D. L., & Landerer, F. W. (2015). Internal and external forcing of sea level variability in the Black Sea. Climate Dynamics, 1-14.

Volkov, D. L., Johns, W. E., & Belonenko, T. V. (2016). Dynamic response of the Black Sea elevation to intraseasonal fluctuations of the Mediterranean sea level. Geophysical Research Letters, 43(1), 283-290.

Yildiz, H., Andersen, O. B., Kilicoglu, A., Simav, M., & Lenk, O. (2008). Sea level variations in the Black Sea for 1993-2007 period from GRACE, altimetry and tide gauge data. Geophysical Research Abstracts, Vol. 10, EGU2008-A-08684

Zatsepin, A. G., Kremenetskiy, V. V., Poyarkov, S. G., Ratner, Y. B., & Stanichny, S. V. (2002). Influence of wind field on water circulation in the Black Sea. Complex Investigation of the Northeastern Black Sea. Nauka, Moscow, 91-105.a

---

## Author Response (AR2)

**Answer to Reviewer 1**

**Reviewer:** Review on "Impact of basin dynamics on the regional sea level trends in the Black Sea" by Kubryakov et al.

This manuscript deals with the relationship between wind curl and circulation in the Black Sea. The successful reconstruction of circulation from the wind without using numerical model is the basic message. As the authors state, the underlying relationships have been known from earlier studies, however they give a good illustration. The idea of the paper is understandable. However, as presented, it needs a major revision. Here are some comments. For the role of the Ekman pumping and its relationship with the distribution of mass, authors could consult also the work of Stanev et al. (2004).

**Answer:** Thank You, we added the reference

**Reviewer:** p. 2, line 20-23. There is text-repetition.

**Answer:** Thank You, we corrected the text

I wonder whether the right title of section 3.1 should be "Sea level trends". What authors describe is the sea level variability with interannual time-scales over-imposed on the global trend.

**Answer:** We have changed the title to "Interannual variability of the Black Sea level"

**Reviewer:** The same comment applies to the title of paper" Impact of basin dynamics on the regional sea level trends in the Black Sea". What authors show, is that the long-term changes of sea level are characterized by certain spatial variations. And that these spatial patterns are reminiscent of, or related to, some circulation patterns. However, when reading the title one could expect to learn how (quantification of mechanisms) the basin dynamics modulate the sea level change.

**Answer:** We agree with the Reviewer and change the title to "Quantifying the impact of basin dynamics on the regional sea level rise in the Black Sea "

P. 7, line 20. Authors could mention that this time lag was estimates by Stanev et al. (2000) as 1-2 months that is they demonstrate a good consistence with previous research in this field based on longer record.

**Answer:** We have added a corresponding sentence and the reference to the text: "The similar time lag (1-2 months) between DSL and wind curl variability was obtained in the earlier study of (Stanev et al., 2000)."

**Reviewer:** P. 8, caption Figure 6. Perhaps better normalize the wind curl by f.

**Answer:** In this study, we decided to use wind curl and not Ekman pumping, because here the wind curl describes only the external forcing. More detailed investigation of the Ekman pumping variability and its connection with the Black sea dynamics was done in our previous study "Kubryakov, A. A., Stanichny, S. V., Zatsepin, A. G., & Kremenetskiy, V. V. (2016). Long-term variations of the Black Sea dynamics and their impact on the marine ecosystem. *Journal of Marine Systems*, *163*, 80-94."

**Reviewer:** P. 9. There is a double use of "6" in Fig. numbers.
**Answer:** Thank You, we have corrected the figure numbering

**Reviewer:** P. 9. Avoid repeating Fig. 1 in Fig. 6a-horizontal pattern.

**Answer:** These figures are very similar, but Fig. 1c shows the total sea level trends, while Fig. 7a shows DSL trends. The figure 7a is shown to demonstrate the magnitude of the DSL trends and highlight the local features of the DSL trends, which are attributed to the mesoscale eddy variability, discussed in this section.

**Reviewer:** P. 9, line 10. The trends in Fig. 6 (b, c, d) are small-scale and patchy. An estimate of their confidence needs to be provided. These figs are not sufficiently considered/explained in the text. Their patchiness makes me think that there is a problem with statistics. Perhaps the whole section "3.3. The impact of mesoscale variability on the sea level trends" can be omitted. Actually, the title of this section is too promising. We see small scale patterns (or noise) but the explanation of how the mesoscale variability impacts the sea level trends is missing (not presented in a convincing quantitative way).

**Answer:** As we see from the figures 1, 6a we can observe a spurious local maximums of the sea level rise in several areas of the basin. These maximums can not be explained by the large-scale Ekman dynamics. Here we use eddy statistics in order to show that at least part of them (e.g. largest maximums in the south-east part) can be explained by the observed changes of the eddies characteristics. Also the methods of automated eddy identification allows to obtain a large statistical array about eddy characteristics, eddy dynamics can be very complex and patchy. Nevertheless we still can observe some more or less high trends of the anticyclone radius and eddy velocity in several places that coincide with the values of the increased sea level. This give

the idea about the effect of long-term eddies variability on the dynamic sea level changes in the basin, that is valuable for this study "Quantifying the impact of basin dynamics on the regional sea level rise in the Black Sea". The detailed investigation of the reason of these changes f eddy dynamics is an interesting and complex task that is out of the scope of our study.

According to Your comments we rewrite this section, redraw the figure 7 and figure 8, slightly extended the text in this section and added a brief explanation of how the mesoscale variability impacts the sea level trends.

[Figure]

Figure 1. Trends (left) and their errors (confidence intervals) (right) for: top - frequency of anticyclones observation, middle - their radius (km), bottom - maximum orbital velocity (m/s).

In Figure 1  we also provide the estimates of the confidence intervals (right panel) for the computed trends (left panel) for frequency of anticyclones observation, their radius (km) and maximum orbital velocity (m/s). As You can see in the most of the areas of the basin the errors are not more than approximately 10% of trend value

**Answer to Editorial comments**

**Editor:** Page 1 Lines 15, 16. I prefer "rise", "fall" (or perhaps "lowering") of sea level, not "increase", "decrease".

There are many later places in the text where this also applies, e.g. page 2 line 1, page 3 line 39 ("falling"), page 12 line 8 ("rising"), page 12 Line 11 ("a rise"), page 12 line 12.

**Answer: Corrected**

**Editor:** Line 32. Better ". . Black Sea MSL then rose at a faster rate . ."

**Answer: Corrected**

**Editor:** Line 14. "reduces" -> "decreases"

**Answer: Corrected**

**Editor:** Line 19. "sea level is rising . ."

**Answer: Corrected**

**Editor:** Lines 20-22. This sentence is a duplicate.

**Answer:Thank You, we corrected the text**

**Editor:** Line 28 and many other places including notation on figures. I prefer "metre" (European spelling) since Napoleon defined it and to avoid ambiguity with a measuring instrument. Also page 3 line 20, figure 2c and its caption, figure 3 caption

**Answer: Corrected**

**Editor:** Line 11. "derived". How? Especially, how do you go from a current vector everywhere to one value which you compare with wind curl.

**Answer:** We have added an explanation to the text:

"The absolute dynamic topography of the Black Sea was computed as the sum of the mapped SLA and a "synthetic" mean dynamic topography of Kubryakov and Stanichny (2011). Surface geostrophic currents (u,v) was computed from absolute dynamic topography the using geostrophic equations: $u_g = -\dfrac{g}{f}\dfrac{\partial h}{\partial y}$ ; $v_g = \dfrac{g}{f}\dfrac{\partial h}{\partial x}$.

Here, $u_g$ and $v_g$ are the zonal component and meridional component of geostrophic velocity; h is the absolute dynamic topography; $f$ is the Coriolis parameter; g is the gravitational acceleration; and x, y are the longitude and latitude, respectively. To describe the basin-scale variability we use the magnitude of geostrophic velocity $U = \sqrt{u^2 + v^2}$ "

**Editor:** Line 18. You need ":" after first "MSL"

**Answer: Thank You, we corrected the text**

**Editor:** Lines 29-30. Better "it rose again"

**Answer: Corrected**

**Editor:** Line 37. "southeastern"

**Answer: Thank You, we corrected the text**

**Editor:** Line 3 (3rd of figure caption). "2017" -> "2007".

**Thank You, we corrected the text**

**Editor:** Line 1. "DSL falls in the centre . ."

**Answer: Corrected**

**Editor:** Figure 3 caption line 3. "isobaths" (spelling).

**Answer:Thank You, we corrected the text**

**Editor:** Lines 5-6. "including most of the north-west shelf"? (Something is missing)

**Answer: This phrase is excluded from the revised version.**

**Editor:** Line 7. Delete "," after "least".

**Answer: Corrected**

**Editor:** Line 18. "mean speed of surface geostrophic currents" c.f. comment on page 3 line 11; how is this derived?

**Answer: We have added an explanation to the text in the Section 2**

**Editor:** Line 19. "rising" -> "increasing" (refers to current speed). "0.5" -> "0.05". "per year".

**Answer: Corrected, Thank You.**

**Editor:** Lines 7-8. Figure 2c (caption) refers to "basin's periphery". Please explain "averaged over the shallow areas" which needs to be something different to find "close agreement".

**Answer:** Thank You, we **have rewritten this part of the text** . **"**The average range of the interannual oscillations of DSL at the first point is about 5 cm, in close agreement with the amplitudes of the DSL averaged along the basin's periphery (depths less than 500 metres) (Fig.2c)"

Page 8.

**Editor:** Figure 5 caption line 4. Needs completion.

**Answer: Corrected**

**Editor:** Line 12. This is not the definition of DSL in section 2.

**Answer: We agree, and we have rewritten this part of the text.**

**Editor:** Line 13. ". . while the total Black-Sea-average sea level (S) trend . ."

**Answer: We have rewritten this part of the text.**

**Editor:** Figure 6b, colour scale needs units.

**Answer: The units of frequency is fraction, so they are dimensionless**

**Editor:** Line 3 and figure 7a. Text and figure disagree. I think the figure is not % but simply fraction of time. C.f. figure 6b.

**Answer: This is right.  We have corrected the text accordingly**.

**Editor:** Figure 9 line 3 (caption line 1). ". . DSL at 33.2°E, 44.7°N reconstructed . ."

**Answer: Corrected**

**Editor:** Line 10. "southeastern"

**Answer: Thank You, corrected**

**Editor:** Line 14. "southeastern"

**Answer: Corrected**

[revised manuscript text omitted]

---

## Author Response (AR3)

**Answer to Editorial comments**

**Thank You for Your valuable comments. Below we provide point-by-point response on Your comments.**

*Editor: Page 1 line 17. "southeastern"? (as elsewhere)*

5 **Answer: Corrected**

*Page 2 line 15. ". . Kubryakova and Korotaev . ."*

**Answer: Corrected**

*Page 3 line 14. "u" and "v" need subscripts "g" as in line 12.*

**Answer: Corrected**

10 *Section 3.1. Page 3 line 34 to page 4 line 2. The overall rate 3.15 mm/year is much less than the combination 26.2 mm/year in 1993-1999 plus -3.0 mm/year in 2000-2007 plus 10.0 mm/year in 2007-2014. There are two reasons for this inconsistency. Firstly, in figure 1b you have a discontinuity in the trend lines between the first two periods. I think this is not acceptable. Secondly, the overall linear trend starts and finishes at different levels from the 3-part fit. This is an inevitable result of a linear trend when*
15 *the variation is not linear. It seems to me that you need to decide what is really the message of this paragraph and to give consistent trends accordingly.*

**Answer: We agree. In order to avoid this problem we decide to exclude the figure 1b from the text and also exclude values of trends for the different periods of time.**

*Page 4 line 1. 1993-1999?*

20 **Answer: Yes, Corrected**

*Page 6*

*Lines 10-11. ". . including the large north-west shelf area. . ."?*

**Answer: Corrected**

*Lines 11-12 and page 7 lines 2-3. These sentences should be merged with page 5 lines 3-5, they are all*
25 *about the same thing.*

**Answer: We corrected the text according to your comments**

*Line 23. Please define ". . mean . . in the basin . ." Do you mean a spatial average of speed (velocity magnitude)? Over the whole area of the Black Sea or the deep part (how deep)?*

**Answer: We changed the phrase on "The basin-averaged speed of surface geostrophic currents $U$ (Fig.4 –**
30 **blue curve) was rising at an approximate rate of 0.05±0.003 cm/s per year, i.e. by 0.3% per year of the average value."**

*Line 24. Better "increasing" not "rising".*

**Answer: Corrected**

*Page 7.*

35 *Figure 4 caption. "basin-averaged" – same question: over the whole area of the Black Sea or the deep part?*

**Answer: Here, we mean over the whole area of the Black Sea**

*Lines 14, 15. "amplitude" is not the same as "range". Range = max – min. Amplitude = max – mean or mean – min. I think you mean "range" in line 14 but you need to check Goryachkin and Ivanov (2006) for line 15.*

5  **Answer: Corrected**

*Page 8*

*Figure 5 caption. Better ". . 33.2°E, 44.8°N . . 35.4°E, 44.0°N . ."*

**Answer: Corrected**

*Lines 13-14. ". . DSL = SLA-MSL*

10  ***Answer: Thank you. We have changed the text on*** DSL=ADT-MSL, where ADT is absolute dynamic topography. Accordingly, we have changed DSL description in the Section 2.

*Page 9*

*Line 6. Page 10 line 3 and figure 8 refer to 38.5°E, 42°N.*

***Answer: Thank you. We have changed the text.***

15  *Line 8. ". . Kubryakov and Stanichny . ."*

**Answer: Corrected**

*Lines 16-17. This sentence repeats page 9 lines 6-7.*

**Answer: Corrected**

*Lines 18-19. Better ". . Positive trends of eddy radius . ."?*

20  **Answer: Corrected**

*Page 10*

*Line 1. Please explain (why the) "Anticyclonic Batumi eddy induces convergent motions"*

**Answer: We add a simple explanation to the text**

*Line 9. Better ". . 31.5°E, 44°N . ."*

25  **Answer: Corrected**

*Figure 8 caption. Better ". . 38.5°E, 42°N . . from . ."*

**Answer: Corrected**

*Editor: Page 1 Lines 15, 16. I prefer "rise", "fall" (or perhaps "lowering") of sea level, not "increase", "decrease".*

30  *There are many later places in the text where this also applies, e.g. page 2 line 1, page 3 line 39 ("falling"), page 12 line 8 ("rising"), page 12 Line 11 ("a rise"), page 12 line 12.*

**Answer: Corrected**

[revised manuscript text omitted]